# Decoding the Proanthocyanins Profile of Italian Red Wines

Panagiotis Arapitsas [1,2,*], Daniele Perenzoni [1], Maurizio Ugliano [3], Davide Slaghenaufi [3], Simone Giacosa [4], Maria Alessandra Paissoni [4], Paola Piombino [5], Elisabetta Pittari [5], Andrea Versari [6], Arianna Ricci [6], Andrea Curioni [7], Matteo Marangon [7] and Fulvio Mattivi [1]

1 Unit of Metabolomics, Research and Innovation Centre, Fondazione Edmund Mach, 38010 San Michele all'Adige, Italy
2 Department of Wine, Vine and Beverage Sciences, School of Food Science, University of West Attica, Egaleo, 12243 Athens, Greece
3 Department of Biotechnology, University of Verona, 37134 Verona, Italy
4 Department of Agricultural, Forest and Food Sciences, University of Torino, 10095 Grugliasco, Italy
5 Department of Agricultural Sciences, Division of Vine and Wine Sciences, University of Napoli Federico II, 83100 Avellino, Italy
6 Department of Agricultural and Food Sciences, University of Bologna, 47521 Cesena, Italy
7 Department of Agronomy, Food, Natural Resources, Animals and Environment (DAFNAE), University of Padova, 35020 Legnaro, Italy
* Correspondence: panagiotis.arapitsas@fmach.it or parapitsas@uniwa.gr

**Abstract:** The Italian wine appellations system is organized in hundreds of origin wines, with unique characteristics that are protected with many denominations of origin. The aim of this work was to analyze and compare the proanthocyanin profile of 12 single-cultivar and single-vintage Italian red wine groups (Aglianico from Campania, Cannonau from Sardinia, Corvina from Veneto, Montepulciano from Abruzzo, Nebbiolo from Piedmont, Nerello Mascalese from Sicily, Primitivo from Apulia, Raboso Piave from Veneto, Sagrantino from Umbria, Sangiovese from Tuscany and Romagna, and Teroldego from Trentino), each one produced in their terroirs under ad hoc legal frameworks to guarantee their quality and origin. All wines were analyzed with a protocol that combined the phloroglucinolysis reaction with an LC-MS/MS instrument. The results underlined Sagrantino wines as the richest in proanthocyanins. Sangiovese, Montepulciano, Nerello, and Teroldego were the richest in B-ring trihydroxylated flavan-3-ols, and especially Nerello was the richest in prodelphinidins. Cannonau, Raboso Piave, Nerello, and Corvina were characterized by C-ring *trans* conformation flavan-3-ols. Nebbiolo and Corvina had high percentages of galloylated flavan-3-ols. Aglianico and Primitivo had the lowest percentages of B-ring trihydroxylated and C-ring trans conformation flavan-3-ols. This information should be useful in better understanding the Italian red wines and valorize them.

**Keywords:** condensed tannins; catechin; epicatechin; galloyl; astringency; *Vitis*; mass spectrometry

## 1. Introduction

Monomeric, oligomeric, and polymeric flavan-3-ols comprise a group of secondary metabolites that has a paramount importance in wine science and industry [1]. They are key elements for the understanding of wine taste and mouthfeel [1–7], color and longevity [1,8–13], taxonomy and authenticity [8,9,14–17], reactivity [1,12], antioxidant properties, and putative health benefits [18–20]. The main flavan-3-ols monomeric forms constitutes catechin, epicatechin, gallocatechin, epigallocatechin, catechin gallate, epicatechin gallate, gallocatechin gallate, and epigallocatechin gallate [21] (Figure 1). The combination of these monomers may deliver an enormous number of different oligomeric and polymeric products, but nature strictly controls their biosynthesis. Grapevine is a good model to investigate flavanols, since there is evidence of a huge quantitative and qualitative diversity of the grape tannins, not only among different cultivars, but also among different

parts of the cluster (e.g., skins, seeds, stalks). Moreover, this diversity is transferred from the fruit into the wine during the process of production of red wines (and white wines with maceration, also known as orange wines), particularly during the skin maceration step, with important effect on the wine quality. However, the biosynthetic mechanisms leading to the formation of oligomers and polymers remains unexplained [22].

catechin: R = H, R1= H
gallocatechin: R = OH, R1= H
catechin gallate: R = H, R1= galloyl
galloatechin gallate: R = OH, R1= galloyl

epicatechin: R = H, R1= H
epigallocatechin: R = OH, R1= H
epicatechin gallate: R = H, R1= galloyl
epigallocatechin gallate: R = OH, R1= galloyl

**Figure 1.** Chemical structures of the major monomeric flavan-3-ols found in grapes and wine.

The biodiversity and enological richness of Italian red wines is one of the results of this natural regulation. In fact, Italy has one of the richest ampelographic heritage that, together with the numerous local terroirs and human needs, over the centuries developed several hundreds of origin wines [23,24]. Nowadays, in Italy, more than 700 k hectares are cultivated with grapevines, coming from over 500 varieties, and producing 118 IGT (Indicazione Geografica Tipica), 330 DOC (Denominazione di Origine Controllata), and 76 DOCG (Denominazione di Origine Controllata e Garantita) wine [23,24]. Even though these origin wines are well distinguished and recognized from a sensory point of view, extensive knowledge concerning their profile at the proanthocyanin level is lacking.

Nowadays, the measurement of free monomeric flavan-3-ols is an easy task, while the study of the oligomeric and—especially—the polymeric fractions is not trivial. Since the analysis of the polymeric forms as single metabolites is problematic, wine scientists and oenologists have adopted various alternative methods. The methods that enable to extract as much information as possible from such polymers are based on their chemical cleavage and depolymerization. In particular, phloroglucinolysis is a degradative method that allows the interflavonoid bond to break down under acidic conditions, with the consequent release of the terminal units as monomers as well as the upper and extension units as C4 phloroglucinol adducts [25–27] (Figure 2).

The outputs of this protocol include: (a) the quantification of the free monomeric flavan-3-ols, present before the depolymerization, (b) the quantification of the monomeric forms released after the reaction, which are the terminal units, (c) the quantification of the upper units, (d) the mean degree of polymerization (mDP) value, (e) information about the galloylation % on the free, terminal, and upper units, (f) data about the B-ring trihydroxylated/dehydroxylated flavanols (B-ring $3\times$OH/$2\times$OH) ratio, of the free, terminal, and upper units, and (g) data about the catechin/epicatechin (*trans/cis*) C-ring structural conformation of the free and terminal units. All these parameters are genetically controlled, and at harvest, these are cultivar-specific, so that they can be used to profile each wine identity and uniqueness. A limit of this protocol is that does not include the A-type oligomers, since the conditions of the assay are optimized for the release of the oligomers having the single linkage (B-Type) [1,25,27–29]. Moreover, while the relative amounts are mainly genetically controlled [22,29–35], the absolute amounts of the single flavan-3-ol might vary due to biotic factors and winemaking techniques [1,34–39].

**Figure 2.** The units of the polymeric flavan-3-ols.

This work was part of a bigger project aiming to study the diversity of the Italian red wines. Previous papers of this project, on the same wines, presented the results concerning the metabolomic space [17], the astringency sensory profile [5,6], the basic oenological parameters, color, and phenolic indices [40], their mid-infrared spectra [41], their terpenoids and norisoprenoids profile [42], and macromolecular profile [43]. The sample set, subject of this project, included 110 Italian red wines, each one produced at industrial scale by a different winery, and divided in 12 groups, including some of the most iconic terroir and cultivar combinations reflecting the diversity of Italian red wines (see materials and methods). The aim of this work was to use a recently developed phloroglucinolysis protocol [25] for the characterization and comparison of monomeric and polymeric flavan-3-ols of this large set of red wines.

## 2. Materials and Methods

The complete sample set included a total of 110 single-cultivar and single vintage red wines. The sample were divided into the following groups: 9 Aglianico wines from the Campania region, 9 Cannonau from Sardinia, 7 Corvina from Veneto, 9 Montepulciano from Abruzzo, 11 Nebbiolo from Piedmont, 3 Nerello Mascalese from Sicily, 11 Primitivo from Apulia, 10 Raboso Piave from Veneto, 10 Sagrantino from Umbria, 7 Sangiovese from Tuscany, 12 Sangiovese from Romagna, and 11 Teroldego from Trentino. All wines were sampled within 3 months after the end of fermentation, in early 2017, from various wineries of the region of origin, and were independently produced according to the standard production practices of each winery. All wine samples were produced by a single cultivar, fermented in stainless-steel vats (without any wood contact); without malolactic fermentation, 50 mg/L of free $SO_2$ was added at the time of sampling/bottling, and wines were bottled in dark glass bottles with Nomacorc Select Bio 500 (Nomacorc, Thimister-Clermont, Belgium) closures. The basic enological analysis of the 110 wines can be found in a previous publication of the same project [40].

The sample preparation for the measurements of the phenolic analytes and the analytical methods used were previously described in detail [25]. For the SPE (Solid Phase Extraction), C18-SPE cartridge (1 g, Waters, Milford, MA, USA) were used, and 10 mL of wine diluted 10 times with $H_2O$ was applied to the cartridge. The 40 mL of methanolic SPE extract were concentrated until dryness, and the sample was then reconstituted in 2 mL of methanol. For the reaction, 100 μL of concentrated and purified wine sample reacted with 100 μL of the phloroglucinol reagent at 50 °C. All the samples were filtered through 0.22 μm polytetrafluoroethylene (PTFE) filters before analysis. The UHPLC-MS/MS analysis was performed on a Waters Acquity UPLC system (Milford, MA, USA), the column used was a

Waters Acquity HSS T3 column 1.8 μm, 150 mm × 2.1 mm (Milford, MA, USA), and the mobile phases was composed of eluent A (0.1% formic acid in water) and eluent B (0.1% formic acid in acetonitrile). The details about the chromatographic multigradient profile, the mass spectrometer parameters, and the method validation information can be found in a previous publication [25].

All chemicals and solvents used were of the highest purity, and pure standard of the three phloroglucinolated compounds were produced and used as external standard for the quantitation, as previously described [25].

The absolute concentrations of the free dimers and monomers (Tables S1 and S2) measured before the phloroglucinolysis and the terminal and upper units (Tables S4 and S6) measured after the phloroglucinolysis were expressed in mg/L. Each terminal unit concentration was calculated by subtracting the corresponding amount of the free monomer, measured before the phloroglucinolysis, from the concentration measured after the phloroglucinolysis. This subtraction allowed us to avoid a systematic error often occuring in this analysis [25]. For the calculation of the mDP, first, all the values were expressed in mol/L, and then, the sum of upper and terminal units was divided by the sum of terminal units. All the formulas for the various calculations are reported in the Supplementary Materials.

The descriptive and ANOVA statistical analysis were made using SPSS V28 (IBM Statistics). For the post hoc multiple comparison of the one-way ANOVA test, Tukey's HSD statistical analysis was performed considering with a $p$-value $< 0.05$.

## 3. Results

The application of a modern UHPLC-MS/MS method gave us the possibility to quantify in an absolute manner up to 15 polyphenols in a fast and sensitive way, and to obtain quantitative information of the monomeric and oligomeric flavan-3-ols profiles of 12 groups of important Italian red wines with certified origin (cultivar-region combination). In the following paragraphs, we will discuss the results for each group separately, apart the two Sangiovese groups, which will be discussed together. The full dataset of the results including the statistical analysis can be found to the Supplementary Materials (Tables S1–S7). Initially, the results will be presented with radar graphs (Figures 3–8), where each parameter is expressed as a percentage of the relative mean value, with 100% being the mean value of 12 groups considered in the study. In this way, it was possible to compare the profiles of the different wine groups by decreasing the influence of the absolute amounts, which are highly influenced by biotic factors (e.g., vintage) and the winemaking process (e.g., length of maceration). The radar graphs include data about (a) the mDP, (b) the total amount of monomers, respectively, in the free, terminal, and upper form, (c) the sum of two major flavan-3-ols, catechin and epicatechin, in their free and terminal forms, (d) the ratio between the C-ring *trans/cis* structural conformations (*trans*: catechin and gallocatechin and *cis*: epicatechin and epigallocatechin), in free and terminal forms, (e) the ratio between the B-ring trihydroxylated/dihydroxylated (B-ring 3×OH/2×OH) derivatives, in the free, terminal, and upper forms, and (f) the % of the galloylated flavan-3-ols, in the free, terminal, and upper form.

### 3.1. Sangiovese

Sangiovese represents probably the most important grape cultivar in Italy, with 54 k cultivated hectares all over the Italian peninsula in 2015 that produce some of the most famous Italian wines (e.g., Brunello di Montalcino and Chianti Classico), including 12 DOCG, 102 DOC, and 99 IGT [24]. This project was focused only on Sangiovese from Tuscany and Romagna, as these are the two major region of Sangiovese production in Italy, and their results were similar. Figure 3 shows a comparison of each Sangiovese group parameters, expressed as a percentage of the mean values. The parameter that characterized Sangiovese, in respect to the other wines, was the high mDP value, with Sangiovese from Romagna having a value of 13.3 and Sangiovese from Tuscany 12.1, considering the overall mDP of all wines was 10.6 (Table 1). The Sangiovese from both regions had similar mean amounts of

free catechin (Sangiovese from Romagna: 15.6 mg/L, Sangiovese from Tuscany: 16.8 mg/L) and free epicatechin (Sangiovese from Romagna: 15.6 mg/L, Sangiovese from Tuscany: 15.9 mg/L) (Table S2). The major upper unit, with about 62%, was epicatechin, while the major terminal unit was catechin (about 50%), with epicatechin being the second (about 32%) (Tables S5 and S7). Sangiovese wines had also a higher percentage of trihydroxylated B-ring flavanol-3-ols in respect the sample set average (Figure 3).

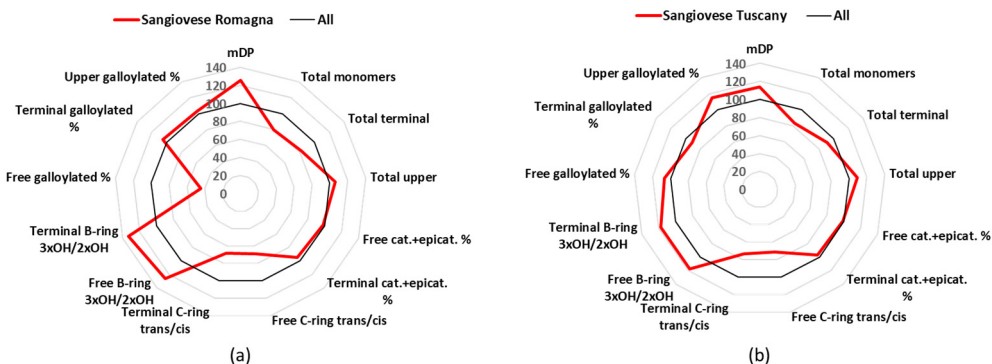

**Figure 3.** Radar graph comparing the percentage of the relative mean values of Sangiovese wines from Romagna (**a**) and from Tuscany (**b**), with the mean values of the 12 groups considered as 100%. cat.: catechin, epicat: epicatechin.

**Table 1.** Basic statistics (mean ± SD) and post-hoc ANOVA of the mDP values.

| Wine Group (Number of Samples per Group) | mDP | ANOVA * |
|---|---|---|
| Sangiovese Romagna (7) | 13.30 ± 2.10 | e |
| Sangiovese Tuscany (12) | 12.11 ± 1.56 | d, e |
| Nebbiolo (11) | 13.21 ± 3.25 | e |
| Aglianico (10) | 9.38 ± 1.66 | a, b, c, d |
| Nerello Mascalese (3) | 11.31 ± 1.99 | c, d, e |
| Primitivo (11) | 8.05 ± 0.80 | a, b |
| Raboso Piave (10) | 10.56 ± 2.90 | a, b, c, d, e |
| Canonnau (9) | 12.01 ± 1.98 | d, e |
| Teroldego (11) | 10.98 ± 1.69 | b, c, d, e |
| Sagrantino (10) | 10.48 ± 1.32 | a, b, c, d, e |
| Montepulciano (9) | 8.50 ± 1.58 | a, b, c |
| Corvina (7) | 7.54 ± 0.91 | a |

* Tukey test. Means that do not share a letter are significantly different (significance value below 0.05).

### 3.2. Montepulciano

Montepulciano is a cultivar mainly found in Abruzzo, cultivated in 27 k hectares (year 2015), and used in 3 DOCG, 36 DOC, and 88 IGT [24]. Montepulciano wines had an mDP mean value (8.5) below the average of all samples (Table 1), free catechin and epicatechin had similar quantities (18.4 and 16.5 mg/L), which were 92% of the total free monomers, and the other free monomers were represented by minor quantities (Tables S2 and S3). The major quantified terminal unit was catechin (55%), while epicatechin represented 30% and gallocatechin 12% (Table S5). For the upper units, it was found that epicatechin represented 69%, epigallocatechin 23%, and epicatechin gallate 7% (Table S7).

### 3.3. Primitivo

Primitivo is cultivated mainly in Puglia, covering 16 k of cultivated hectares in Italy (year 2015), and producing 1 DOCG, 8 DOC, and 51 IGT [24]. The analyzed Primitivo group demonstrated one of the lowest mDP values (8.1), but its most peculiar characteristic was the low percentage of flavan-3-ols with three -OH in the B-ring (Figure 4b, Table 1). Catechin and epicatechin, with 19.3 and 18.2 mg/L, represented 97% of the free monomers,

while the mean values for gallocatechin (0.1 mg/L) and epigallocatechin (1.1 mg/L) were the lowest registered (Tables S2 and S3). The same situation was found for the terminal units, with the monomers having 2 -OH on the B-ring to be the 95%, while concerning the upper units, the trihydroxylated-flavan-3-ols represented only 15% (Tables S4–S7).

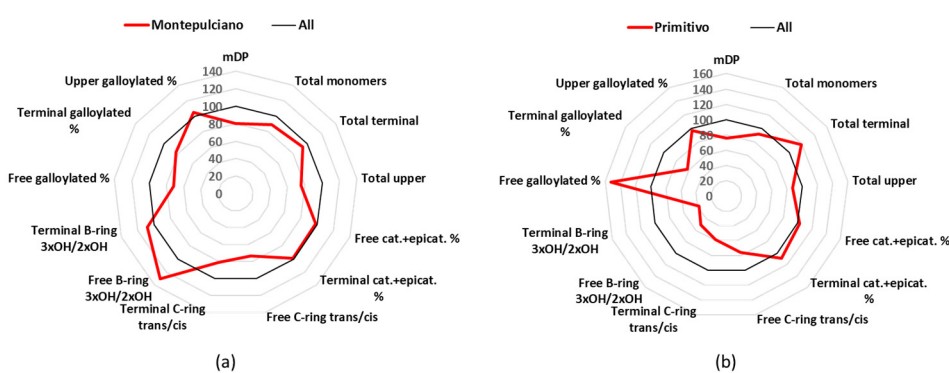

**Figure 4.** Radar graph comparing the percentage of the relative mean values of Montepulciano (**a**) and Primitivo (**b**), with the mean values of the 12 groups considered as 100%. cat.: catechin, epicat: epicatechin.

### 3.4. Aglianico

Aglianico wines are produced in south Italy (mainly Campania and Basilicata), from vines that are cultivated in 9 k hectares, and produce 3 DOCG, 12 DOC, and 66 IGT [24]. Aglianico wines had an mDP of 9.8 (Table 1), which is lower than the average of all analyzed wines and were characterized by small level of trihydroxylated- B-ring flavan-3-ols and high concentration of monomeric flavan-3-ols (Figure 5a). The total concentration of free monomers was 72.6 mg/L that was statistically significant higher in respect to all the other groups (Table S2). In total, 97% of the free monomers was catechin (35.4 mg/L) and epicatechin (35.3 mg/L) (Table S3). The B-ring trihydroxylated flavan-3-ols represented a minor part of the terminal (5.5%) and upper (13%) units.

### 3.5. Corvina

Corvina is a cultivar used to produce Amarone di Valpolicella, Valpolicella and other famous Italian wines, so it is found mainly in Veneto; there are about 7 k of cultivated hectares in Italy, which participate in the production of 3 DOCG, 4 DOC, and 14 IGT [24]. Major characteristic of the Corvina samples was the lowest mDP mean value, 7.5, a difference that was statistically significant for 6 of the 11 other analyzed cultivars (Table 1). Other characteristics are a prevalence of dihydroxylated B-rings flavan-3-ols (catechin and epicatechin are the 96% of the free monomers) and that most of them have the *cis*-configuration of catechin (Figure 5b, Table S3). For example, free catechin (27.4 mg/L or 61%) is almost doubled in respect to the epicatechin (15.8 mg/L or 34%); additionally, the terminals catechin (25.5 mg/L or 67%) and gallocatechin (3.4 mg/L or 10%) together are almost three times higher than epicatechin (9.5 mg/L or 21%) and epigallocatechin (0.6 mg/L or 1%) together (Tables S2–S5). As far as the upper units is concerning, Corvina had generally the lowest concentrations, with the mean amount of epicatechin-Phl (229.2 mg/L) being statistically significant against all other cultivars, epigallocatechin-Phl (68.2 mg/L) being statistically significant different against 8 other cultivars, and epicatechin-gallate-Phl (16.0 mg/L) being statistically significant different against 6 other cultivars (Table S6).

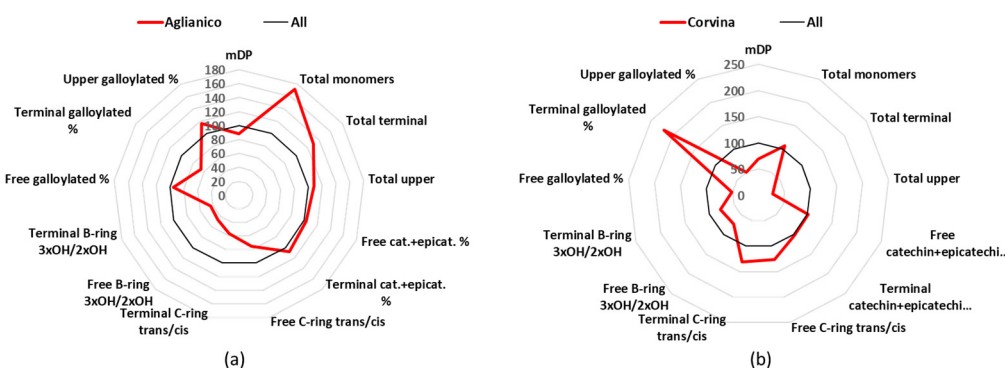

**Figure 5.** Radar graph comparing the percentage of the relative mean values of Aglianico (**a**) and Corvina (**b**), with the mean values of the 12 groups considered as 100%. cat.: catechin, epicat: epicatechin.

### 3.6. Cannonau

Cannonau is a cultivar of the Sardinia Island, and it covers 6 k cultivated hectares in Italy, to produce 2 DOC and 15 IGT [24]. The mean value of the mDP Cannonau group was 12.0, which was higher than the average value of all the 12 groups (Table 1), but the most interesting result concerned the catechin/epicatechin conformation ratio (Figure 6a). Catechin and epicatechin represented 95% of the free monomers, but catechin alone corresponded to 60% and epicatechin only 35% (Table S3). Catechin and gallocatechin together were 78% of the terminal units (63 + 15%) (Table S5).

### 3.7. Nebbiolo

Nebbiolo is an important and famous Italian cultivar, and although in Italy it is cultivated in 6 k hectares, it produces world-famous wines such as Barolo and Barbaresco, and it is used in 7 DOCG, 22 DOC, and 36 IGT [24]. The Nebbiolo wines in this study showed the second highest mDP value with a mean of 13.21 (Table 1). However, the most interesting characteristic of the Nebbiolo wines was the high concentration in galloylated flavan-3-ols (Figure 6b). The mean value for the free catechin gallate was 0.2 mg/L, thus being the highest one and statistically significant different against 10 of the other wine groups (Table S2). If we take in consideration the percentage values, catechin gallate covered 0.5% of all the free monomers, which again was the highest value between all the cultivars and statistically significant different against 10 of the other wines groups (Table S3). Nebbiolo was also in the group of samples that were characterized by the catechin *trans*-conformation rather than the epicatechin *cis*-conformation. For example, free catechin was 60% of the free monomers and epicatechin 35.6% (Table S3). In the case of the terminal units, catechin was 60%, epicatechin 27%, gallocatechin 11%, and epigallocatechin 2% (Table S5).

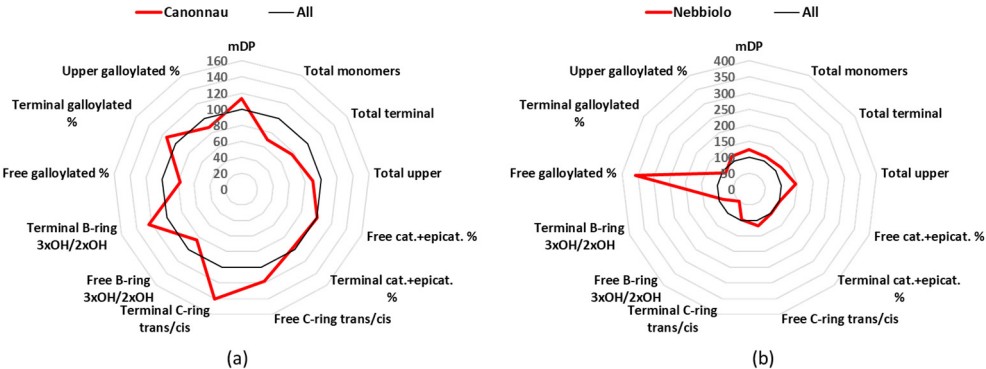

**Figure 6.** Radar graph comparing the percentage of the relative mean values of Cannonau (**a**) and Nebbiolo (**b**), with the mean values of the 12 groups considered as 100%. cat.: catechin, epicat: epicatechin.

### 3.8. Nerello Mascalese

Nerello Mascalese is a minor cultivar of Sicily in term of cultivated surface, with about 2.9 k hectares of production in Italy, and produces 9 DOC and 16 IGT wines [24]. The mDP of the Nerello Mascalese group (11.31) was close to the average value (Table 1). Catechin was the main free monomer (70%), with 27.3 mg/L as the mean value, epicatechin concentrations were very low (9.1 mg/L and 23% of the free monomers), and gallocatechin showed the statistically significant highest concentration (0.9 mg/L and 2% of the free monomers) of all the wine groups (Tables S2 and S3). The same trend was also found in the terminal units, since catechin represented 58%, epicatechin 16% (the lowest and statistically significant among seven wine groups), and gallocatechin 22%, which was statistically the highest value between all the wines (Figure 7a, Tables S4 and S5). For the terminal units, Nerello Mascalese wines showed the highest percentage of epigallocatechin (35%) and the lowest of epicatechin (59%) (Table S5).

### 3.9. Sagrantino

Sagrantino is a cultivar mainly cultivated in Umbria, which covers about 0.9 k hectares in Italy, and produces 1 DOCG, 1 DOC, and 14 IGT wines [24]. These wines had a mDP close to the average one, namely 10.5 (Table 1), and they were characterized by a high percentage of free dihydroxylated B-ring (96%), with free catechin (24.0 mg/L and 50% of the free monomers) and epicatechin (22.1 mg/L and 46% of the free monomers) having analogous mean concentration (Tables S2 and S3). However, in the terminal units catechin was the major one with 67%, followed by epicatechin (22%) and gallocatechin (9%) (Table S5). Considering the absolute values of the terminal units, Sagrantino wines statistically had the highest amount of catechin, with a mean value (123,4 mg/L) that was double the average value of all samples (Table S4). The Sagrantino group also statistically had the highest amount of total upper units, i.e., 2565 mg/L (Figure 7b, Tables S6 and S7).

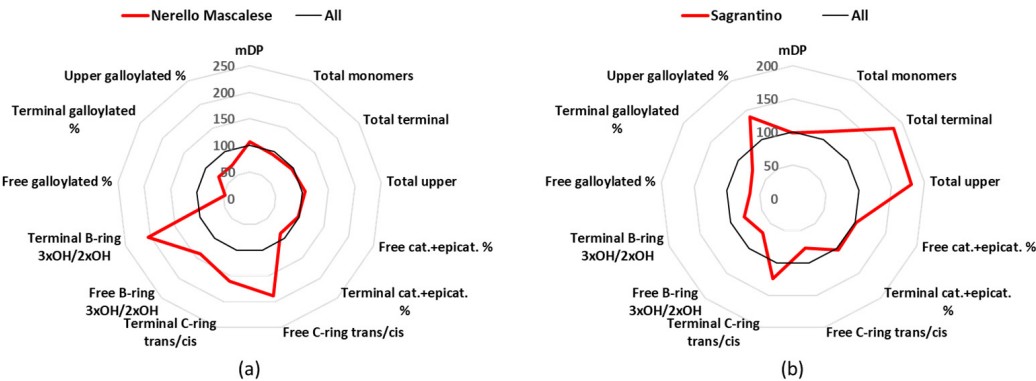

**Figure 7.** Radar graph comparing the percentage of the relative mean values of Nerello Mascalese (**a**) and Sagrantino (**b**), with the mean values of the 12 groups considered as 100%. cat.: catechin, epicat: epicatechin.

### 3.10. Teroldego

Teroldego is a cultivar of a northern Italian region, Trentino, which is cultivated in 0.6 k hectares, and gives 4 DOC and 45 IGT wines [24]. The mDP values for the Teroldego wines were close to the average (Table 1), and their main characteristic was the high percentage of free epigallocatechin (Table S3). Catechin and epicatechin together made up 89% of the free monomers, this being the lowest value, while epigallocatechin with 4.8 mg/L covered 11% (Table S3). These two values of epigallocatechin for Teroldego wines were the highest among all the groups, and statistically significant. For the terminal units, catechin was 50%, epicatechin 38%, and gallocatechin 10%; for the upper units, epicatechin represented 65%, epigallocatechin 27%, and epicatechin gallate 8% (Figure 8a and Table S5).

### 3.11. Raboso Piave

Raboso Piave is mainly found in Veneto (Piave area), is grown in about 0.5 k hectares, and produces 2 DOCG, 7 DOC, and 8 IGT wines [24]. The Raboso Piave group was part of the wines that had a mDP value close to the average (10.2) (Table 1), and were richer in *trans*-conformation analogues. Catechin (19.0 mg/L) represented 63% of the free monomers and epicatechin (9.1 mg/L) 30% (Tables S2 and S3). The situation was similar for the terminal units, with catechin comprising 67% and epicatechin 18% (Figure 8b and Table S5).

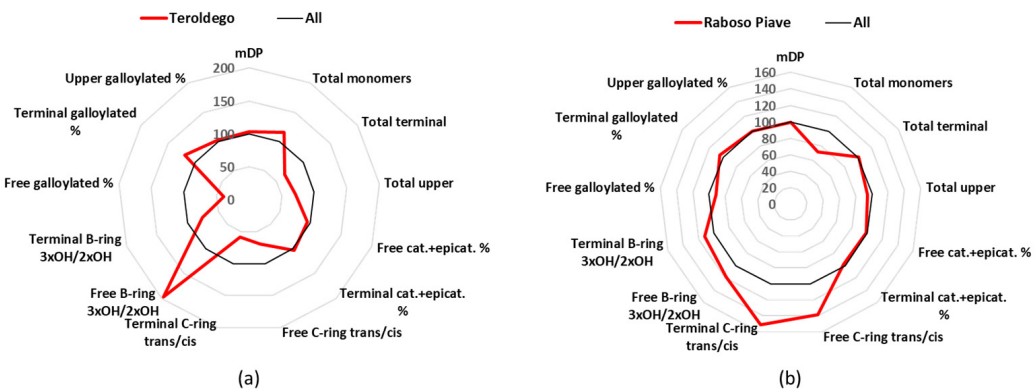

**Figure 8.** Radar graph comparing the percentage of the relative mean values of Teroldego (**a**) and Raboso Piave (**b**), with the mean values of the 12 groups considered as 100%. cat.: catechin, epicat: epicatechin.

## 4. Discussion

Having such a well-defined sample set of 110 wines, which covered a large part of the Italian enological red wine heritage, offered the unique possibility to extract several information concerning their comparison, and to describe, in molecular terms, the tannins' diversity for each combination cultivar and terroir. In what concerns the absolute concentration, Aglianico appears to have the highest amount of free monomers (statistically significant against all the groups) with a mean of 73 mg/L, but Sagrantino had the highest total mean concentration (statistically significant against 10/11 groups) for both terminal (187 mg/L) and upper units (2565 mg/L). This finding is in agreement with previous works in which Sagrantino was demonstrated to be the richest cultivar in flavan-3-ols when compared with other Italian cultivars [25,44]. On the other hand, Corvina wines showed the lowest total mean concentration for both terminal (39 mg/L) and upper units (384 mg/L) (Tables S2–S7). This agrees with the phenolic data discussed in a previous publication based on the same sample set, where Sagrantino had the highest and Corvina the lowest values, for the Folin Ciocalteu, Proanthocyanidin, and Vanillin assays [40].

In wine science, it is well known that tannins bind with proteins, forming aggregates that, together with the polysaccharides, result in the formation of wine colloids [45]. Trying to combine the data of this work with previous results on protein content [43], it seems possible to explain the much high concentration of Corvina in proteins (about 110 mg/L), with respect to all the other wine groups, since these wines were the poorest in flavan-3-ols. However, the same reasoning cannot be applied to other wines. For example, Sagrantino was one of the wine groups with an average protein concentration, while Montepulciano or Primitivo wines, which had low amounts of proteins, did not contain a high amount of flavan-3-ols. Therefore, it does not seem possible to make simple conclusions about a possible inverse correlation between the content in proteins and flavan-3-ols of a wine. However, it has been hypothesized that wines can be better differentiated based on the ratio of their total protein content with their content in tannins reactive to BSA [31], which are the tannins able to interact with wine proteins to form colloidal subunits [45]. Indeed, the authors suggested that

the type of tannins present in a wine, a varietal feature, is likely to be the driver of the formation of protein-tannin aggregates in red wines, a fact that could lead to the formation of tannin-protein colloidal complexes differing in size and stability.

A very interesting characteristic of the Sangiovese wines was that, although all the values were grouped around the average, they had the highest mDP values (12–13) (Table 1), which is a known characteristic for Sangiovese wines [25]. These high mDP values did not corresponded to the highest perceived astringency as measured for Sagrantino and Nebbiolo (highest drying and lowest velvet), but similarly to these wines, Sangiovese wines were correlated with bitter taste and strong astringency sub-qualities (harsh, drying, and dynamic), highlighted as discriminating sensory variables of the three wines compared to the others [5,6], showing that astringency is a very complex sensation to explain chemically [4,33,46–48]. It has been previously underlined that the structural characteristics of the flavan-3-ols play important role in astringency perception. Indeed, it was demonstrated that catechin is less bitter than epicatechin when tested at the same concentration [2]. Additionally, galloylation is positively correlated to the "drying" astringent character, and epigallocatechin units tend to lower the "coarse" astringent perception [4]. Therefore, it is important to further explore the flavan-3-ols' structural characteristic of the wine groups, subject of this work. The wine groups could be divided into two major categories, the first one will include the wines with an equivalent concentration of catechin and epicatechin (Aglianico, Montepulciano, Primitivo, Sangiovese, Sagrantino, and Teroldego), and a second where catechin, which have a *trans* conformation in the C-ring, was predominant (Cannonau, Corvina, Raboso Piave, Nebbiolo, and Nerello Mascalese) (Figure 9 and Tables S2–S7). Catechin is known for being less bitter that epicatechin [2], and for being less reactive in the C-ring [12,49]. In fact, most of the polyphenols produced, during winemaking and wine aging, between flavan-3-ols and acetaldehyde (e.g., ethyl bridge anthocyanin-flavanols or flavanols-flavanols) or $SO_2$ ($4\beta$-sulfonated-flavanols) contain epicatechin [12,49–51].

In addition, the sum of the two B-ring dihydroxylated flavan-3-ols, catechin and epicatechin, for some wines represented more than 96% of the total (Nebbiolo, Aglianico, and Primitivo), while others were richer in B-ring trihydroxylated flavan-3-ols (Nerello Mascalese, Montepulciano, Sangiovese, and Teroldego), and catechin with epicatechin together comprised less than 93% (Figure 9 and Tables S2–S7). This issue is of great importance for the upper units, where Nerello Mascalese (35%), Sangiovese (30%), Teroldego (27%), and Raboso Piave (26%) are very rich in epigallocatechin, and they also have mDP values above the average. According to Vidal et al. [4], epigallocatechin units tend to lower the "coarse" perception of the wine. In general, as the number of the hydroxyl groups on the flavan-3-ols B-ring increases, the antioxidant activity of the flavan-3-ol increases. Therefore, the trihydroxylated flavan-3-ols are consider better antioxidants in respect to the mono- and dihydroxylated [52,53].

The study of Vidal also underlined that a "drying" astringent character is positively correlated the degree of galloylation [4]. Sagrantino, Nebbiolo, and Aglianico where the wine groups with the highest percentage values of galloylated polymers, and the first two perceived as the most "drying" wines [6]. Of course, in the overall astringency evaluation, it is important to take in consideration the mDP as well and this value indicates that only Nebbiolo had a mean mDP higher than the group's average (Table 1). Nebbiolo is known to produce wines with both high acidity and proanthocyanidin amounts when young, and for that, they require long ageing to reach a balanced taste [54].

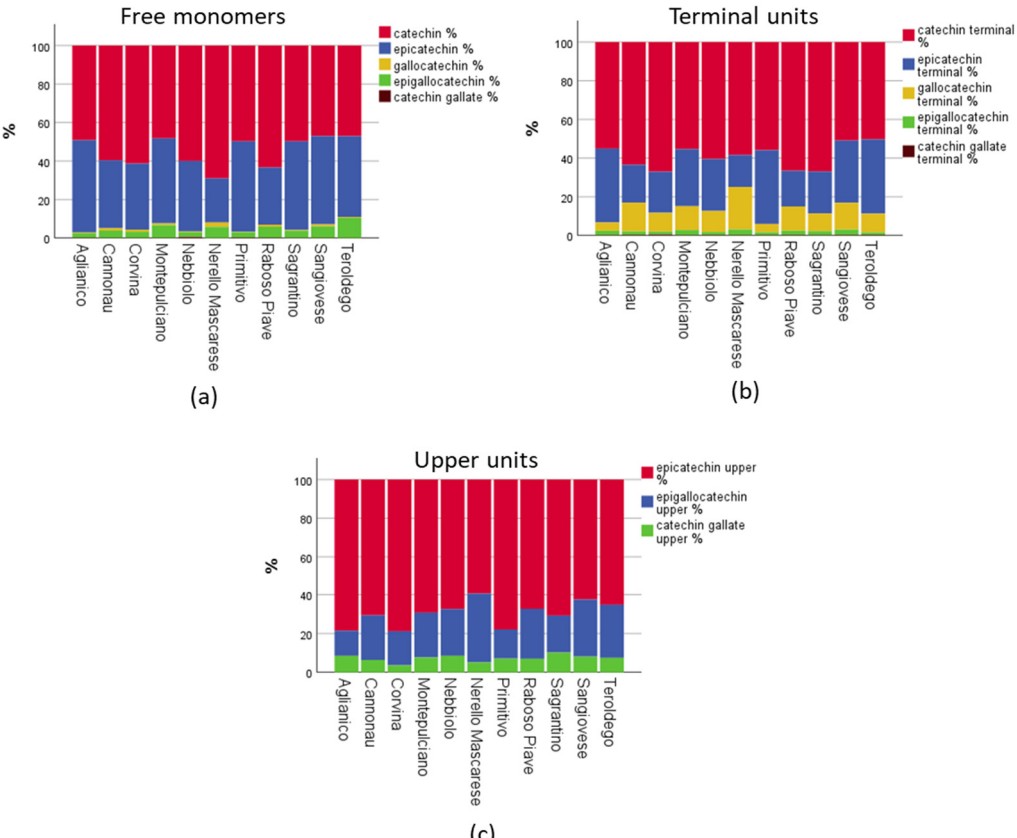

**Figure 9.** Relative abundance of the quantified flavan-3-os between the (**a**) free monomers, (**b**) terminal units, and (**c**) upper units.

## 5. Conclusions

The results of this work enriched a dataset based on the chemical, physical, enological, and sensorial analysis of 110 well-defined single-cultivar Italian red wines. The added information provides a detailed and quantitative image of the most abundant class of polyphenols, i.e., the monomeric, oligomeric, and polymeric flavan-3-ols, key factors to consider when trying to optimize the winemaking procedure to obtain wines of higher quality. In the quest to understand and explain several wine quality characteristics, such as mouthfeel, color, and longevity, the comparison of the composition of a set of very diverse red wines, representing different prototypes of quality, can be of great practical importance.

Specifically, the outputs of the study indicated that before phloroglucinolysis and in absolute values, Aglianico was the richest group in monomeric and dimeric flavan-3-ols, while after the phloroglucinolysis, it was the Sagrantino the group that showed the highest total flavan-3-ols amount. Nerello Mascalese group showed the highest absolute values in polymeric trihydroxylated flavan-3-ols, indicating that it is variety rich in prodelphinidins. Sangiovese and Nebbiolo had the highest mDP values, and Corvina the lowest. Aglianico and Primitivo had the lowest percentages of B-ring trixydroxylated and C-ring trans conformation flavan-3-ols. The groups of Sangiovese, Montepulciano, Nerello Mascalese, and Teroldego were characterized by B-ring trihydroxylated flavan-3-ols (e.g., gallocatechin and epigallocatechin), and Nebbiolo and Corvina were the richest in galloylated analogues.

Moreover, the results highlighted the variability and biodiversity of the Italian enological products. While other key parameters—such as the content in sugars, acids, or minerals—can be interpreted in the same direction for every red wine, the diversity in both the amount and composition of the tannins suggest that the enologist should have a specific range of reference values, for the composition of tannins specific for each cultivar. The final dataset, or these outputs alone, can provide researchers and producers key elements related to all the production steps, from the vineyard to the consumer.

**Supplementary Materials:** The following supporting information can be downloaded at: https://www.mdpi.com/article/10.3390/beverages8040076/s1, Table S1: Basic statistics and ANOVA analysis of the dimeric procyanidins; Table S2. Basic statistics and ANOVA analysis of the free monomers' absolute values; Table S3. Basic statistics and ANOVA analysis of the free monomers' percentage values; Table S4. Basic statistics and ANOVA analysis of the terminal units' absolute values; Table S5. Basic statistics and ANOVA analysis of the terminal units' percentage values; Table S6. Basic statistics and ANOVA analysis of the upper units' absolute values; Table S7. Basic statistics and ANOVA analysis of the upper units' percentage values; Formulas for the calculations. Refs. [1,4,16,22,25,29–35] are cited in Supplementary Materials.

**Author Contributions:** Conceptualization, all; methodology, F.M., D.P. and P.A.; validation, D.P. and P.A.; formal analysis, D.P. and P.A.; resources, all; data curation, F.M., D.P. and P.A.; writing—original draft preparation, F.M., D.P. and P.A.; writing—review and editing, F.M., D.P., P.A., D.S., S.G., P.P., A.V., M.A.P., E.P., A.R. and M.M.; visualization, P.A.; supervision, F.M. and P.A.; project administration, F.M.; funding acquisition, F.M., M.U., S.G., A.V., A.C. and P.P. All authors have read and agreed to the published version of the manuscript.

**Funding:** This work was supported by the Italian Ministero dell'Istruzione, Università e Ricerca (MIUR) project PRIN 20157RN44Y.

**Data Availability Statement:** All data can be found in the Supplementary Materials.

**Acknowledgments:** The authors would like to thank the Italian wineries that provided wine samples for this study, and the other members of D-Wines project: A. Gambuti, V. Gerbi, L. Rolle, L. Moio, G. P. Parpinello, A. Rinaldi, S. Río Segade, B. Simonato, G. Tornielli, and S. Vincenzi.

**Conflicts of Interest:** The authors declare no conflict of interest.

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
