# Peer review of "Decoding the Proanthocyanins Profile of Italian Red Wines"

_beverages, doi:10.3390/beverages8040076_

Round 1
Reviewer 1 Report
The study on the composition of tannin between Italian varietal wines is interesting. However, in this study, the fact that each variety comes from a different appellation (except Sangiovese) limited the exploration of terroir. Therefore, this study can only report on the tannin composition of each wine and lacks comparative studies.
1. Introduction.
Two main topics, one is Italy and the other is tannins. An introduction to Italian appellations and varieties should be in the introduction. If there is too much content, the main points should be summarised. l.79-89, source? l,55 orange wine?
2. Materials and Methods.
It is strongly urged to complete the Materials and Methods. this study focuses on tannins and the determination and analysis of the various subunits of tannins. Therefore, it is necessary to write down formulas for each indicator and to indicate whether they are calculated using concentration or peak area. Otherwise, the data in the results will be unreliable. Especially for Terminal B-ring 3xOH/2xOH, Free B-ring 3xOH/2xOH, etc.
3. Results.
Please consider carefully the data analysis techniques used. It is incorrect that all the Figures in the main text are of little effect and that the analysis and data are almost sourced from Supplementary Tables. The data are not visible because the supplementary tables cannot be downloaded, however, the data from 12 varieties (11 origins) is not a huge data set (if presented as a mean). Therefore, a better presentation is needed. The standard deviation is not visible with the radar plot; moreover, the Sangiovese variety does not show significant differences between the two appellations.
4. Conclusions.
This part is missing important conclusions from this study. Rewrite.
Author Response
Beverage reviewer 1
The study on the composition of tannin between Italian varietal wines is interesting. However, in this study, the fact that each variety comes from a different appellation (except Sangiovese) limited the exploration of terroir. Therefore, this study can only report on the tannin composition of each wine and lacks comparative studies.
We would like to thank the reviewer for his/her feedback and the valuables comments. In fact, aim of this project was not the study of monovarietal wines produced from grapes coming from the same microclimate and produced by the same protocol, or to study wines of the same cultivar but originated from different regions/microclimates, or to study the effect of different viticultural or oenological techniques. As we pointed out in all the publications of this project, the aim was to study the actual biodiversity of the Italian red origin wine production, by covering most of the Italian territory and including most of the iconic terroir and cultivar combinations, and therefore also by including the real production protocol variability. Since in the literature can be found studies where subsets of the selected varieties were cultivated under the same protocol and in the same terroir, we also had the possibility to compare if the outcomes of these previous studies are valid also at the real industrial production. Therefore, this study provides new information on the proanthocyanidin profile, give an image about the Italian oenological variability, and help us to understand their quality and sensorial differentiation. We believe that such studies are missing from the literate, and that can provide interesting information to the scientists, the producers, and the consumers.
- Introduction.
Two main topics, one is Italy and the other is tannins. An introduction to Italian appellations and varieties should be in the introduction. If there is too much content, the main points should be summarised.
The reviewer is correct that this information was missing, but initially we did not include them because they can be found in the previous papers related to the project. However, by considering this suggestion, the following general information on the Italian appellations have been added:
Nowadays, in Italy more than 700k hectares are cultivated with grapevines, coming from over 500 varieties, and producing 118 IGT (Indicazione Geografica Tipica), 333 DOC (Denominazione di Origine Controllata), and 74 DOCG (Denominazione di Origine Controllata e Garantita) wines.
l.79-89, source?
We added references for all the topics included in the paragraph.
l,55 orange wine?
Taking in consideration the OIV we changed “orange wine” with “white wines with maceration, also known as orange wines”
- Materials and Methods.
It is strongly urged to complete the Materials and Methods. this study focuses on tannins and the determination and analysis of the various subunits of tannins. Therefore, it is necessary to write down formulas for each indicator and to indicate whether they are calculated using concentration or peak area. Otherwise, the data in the results will be unreliable. Especially for Terminal B-ring 3xOH/2xOH, Free B-ring 3xOH/2xOH, etc.
We apologise for not being very clear at this point. All the formulas are now included in the Supplementary Material and the following paragraph was added in the main text:
The absolute concentrations of the free dimers and monomers (Tables S1-S2) measured before the phloroglucinolysis, and the terminal and upper units (Tables S4 and S6) measured after the phloroglucinolysis, were expressed in mg/L. Each terminal unit concentration was calculated by subtracting the corresponding amount of the free monomer, measured before the phloroglucinolysis, from the concentration measured after the phloroglucinolysis. This subtraction allowed us to avoid a systematic error often occur in this analysis. For the calculation of the mDP, first all the values were expressed in mol/L and then the sum of upper and terminal units was divided by the sum of terminal units. All the formulas for the various calculations are reported in the Supplementary Materials.
- Results.
Please consider carefully the data analysis techniques used. It is incorrect that all the Figures in the main text are of little effect and that the analysis and data are almost sourced from Supplementary Tables. The data are not visible because the supplementary tables cannot be downloaded, however, the data from 12 varieties (11 origins) is not a huge data set (if presented as a mean). Therefore, a better presentation is needed. The standard deviation is not visible with the radar plot; moreover, the Sangiovese variety does not show significant differences between the two appellations.
We really apologize by the fact that the reviewer could not review the Supplementary Material too. We don’t understand where the problem was, but we hope that the reviewer will have access to the Supplementary Materials now. The Tables with the all the data are too big to include them in the main text. Already the main text has 9 Figures, and the import of all the Tables from the Supplementary Material (8 Tables) will make the manuscript too heavy. However, we decide to include in the main text the Table with the mDP values. The full data, with the statistical analysis and the standard deviation can be found in the Supplementary Material. The reviewer can see that the data set is rather big, since for the 12 wine groups have 18 absolute values (with their standard deviation and Anova analysis), 13 relative values in % (with their standard deviation and Anova analysis) and the mDP values table. All the readers will have the possibility to view the results in detail, and the spider plots will provide the best way for the fast and effective visualisation of such e big data set.
- Conclusions.
This part is missing important conclusions from this study. Rewrite.
The following paragraph was added:
In specific, the outputs of the study indicated that before phloroglucinolysis and in absolute values, Aglianico was the richest group in monomeric and dimeric fla-van-3-ols, while after the phloroglucinolysis it was Sagrantino the group showing the highest total flavan-3-ols amounts. Nerello Mascalese group showed the highest absolute values in polymeric trihydroxylated flavan-3-ols, indicating that is a variety rich in prodelphinidins. Sangiovese and Nebbiolo had the highest mDP values, and Corvina the lowest. Aglianico and Primitivo had the lowest percentages of B-ring trixydroxylated and C-ring trans conformation flavan-3-ols. The groups of Sangiovese, Montepulciano, Nerello Mascalese, and Teroldego were characterized by B-ring trihydroxylated flavan-3-ols (e.g., gallocatechin and epigallocatechin), and Nebbiolo and Corvina were the richest in galloylated analogues.
Reviewer 2 Report
Decoding the proanthocyanins profile of Italian red wines
The objective of this study is very well established, and the methodology proposed appropriate to achieve the research goals.
Proanthocyanins are the main polyphenols of red wines and are determinant of their quality. As their contents may change a lot according to agronomic and climate variables, their profiles are strongly related to genetic factors (as shown in Figure 3 for Sangiovese Romagna vs Sangiovese Tuscany). Thus, studying the proanthocyanin characteristic profile of some of the grape cultivars that stand out among the great diversity of varieties offered by Italian viticulture is of great importance for their characterization and to help understand the relationship between tannin composition and the characteristics of wines.
I understand that this work is part of a bigger project aiming to study the diversity of the Italian red wines. Thus, after carefully reading the manuscript, I think this investigation makes a great contribution in the achievement that main goal.
That said, I have a particular remark. The wine proanthocyanins profiles may be greatly affected by the relative extraction to wine of flavanols from seeds and skin. This may greatly depend on the maceration time. It is stated in the manuscript that the wines “were independently produced according to the standard production practices of each winery”. However, I wonder if there are winemaking procedures characteristic of each region of production or each grape cultivar that imply substantial differences in maceration times that could be biasing the results. If it is the case, there is still a lot of value in this study, but the fact should be mentioned and discussed. The discussion of this fact must be present in the manuscript.
Specific comments:
L103: it would be better to establish in a separate sentence, that the research was made over a total of 110 wines.
L 327-342: It is interesting to discuss the results of the flavan-3-ol of the wines relative to the typical contents of proteins found in previous works for each cultivar. Nevertheless, the other factors on which the wine flavanol concentrations depended (even when not all has been clearly elucidated yet), particularly the synthesis potential of each cultivar (even if it was not measured in this research), cannot be omitted of the discussion.
L 346-348: This paragraph should be revised and rewritten to make it more meaningful.
L 349-: This paragraph should begin to make a nexus with the previous discussion. In the way it is, it seems unconnected of the previous discussion.
L 355-357: A reference for this statement?
L 355-368: It would be better to begin the sentence in a different way to better integrate it with the context.
I consider that the Conclusions section could be improved with more factual statements drawn from the extensive and valuable results obtained by the authors.
Author Response
Reviewer 2
The objective of this study is very well established, and the methodology proposed appropriate to achieve the research goals.
Proanthocyanins are the main polyphenols of red wines and are determinant of their quality. As their contents may change a lot according to agronomic and climate variables, their profiles are strongly related to genetic factors (as shown in Figure 3 for Sangiovese Romagna vs Sangiovese Tuscany). Thus, studying the proanthocyanin characteristic profile of some of the grape cultivars that stand out among the great diversity of varieties offered by Italian viticulture is of great importance for their characterization and to help understand the relationship between tannin composition and the characteristics of wines.
I understand that this work is part of a bigger project aiming to study the diversity of the Italian red wines. Thus, after carefully reading the manuscript, I think this investigation makes a great contribution in the achievement that main goal.
That said, I have a particular remark. The wine proanthocyanins profiles may be greatly affected by the relative extraction to wine of flavanols from seeds and skin. This may greatly depend on the maceration time. It is stated in the manuscript that the wines “were independently produced according to the standard production practices of each winery”. However, I wonder if there are winemaking procedures characteristic of each region of production or each grape cultivar that imply substantial differences in maceration times that could be biasing the results. If it is the case, there is still a lot of value in this study, but the fact should be mentioned and discussed. The discussion of this fact must be present in the manuscript.
We absolutely understand the comment of the reviewer. As we wrote in the manuscript, we collected all the samples within three months after the end of the alcoholic fermentation from various wineries, and the idea was to include the maximum variability possible. Therefore, each winemaker applied the usual production protocol of the winery, in order to produce a typical red wine and according to the specific production disciplinary. The absolute flavan-3-ols’ concentration of the analysed samples may vary due to oenological practises and the vintage, but the profile for each cultivar is mainly genetically controlled. That was one of the main reasons why the results are discussed more are relative % concentrations rather than absolute amounts. The various production regulations, of the selected single-cultivar wines, do not specify substantial differences for the winemaking operations as they refer to local, consolidated production techniques for still red wines. The specification always variable among regulations is regarding the mandatory aging period in oak casks, however this factor cannot influence the findings of this study because the wines were sampled before oak contact (see Materials and methods section). However, the idea of the project was to study the actual variability of the Italian red wines as these are produced.
We tried to underline this issue better in the first paragraph of the Results section of the revised version.
Specific comments:
L103: it would be better to establish in a separate sentence, that the research was made over a total of 110 wines.
Done
L 327-342: It is interesting to discuss the results of the flavan-3-ol of the wines relative to the typical contents of proteins found in previous works for each cultivar. Nevertheless, the other factors on which the wine flavanol concentrations depended (even when not all has been clearly elucidated yet), particularly the synthesis potential of each cultivar (even if it was not measured in this research), cannot be omitted of the discussion.
We included the following phrase in the Introduction:
Moreover, while the relative amounts are mainly genetically controlled [22,31–36], the absolute amounts of the single flavan-3-ol might vary due to biotic factors and wine-making techniques [1,35,37–40].
We preferred to include this phrase in the Introduction and not in the Discussion, since the specific project did not include genetic measurement due to the experimental design.
L 346-348: This paragraph should be revised and rewritten to make it more meaningful.
This paragraph was enriched to be clearer.
Very interesting characteristic of the Sangiovese wines was that, although all the values were grouped around the average, they had the highest mDP values (12-13) (Table 1), which is a known characteristic for Sangiovese wines [25]. These high mDP values didn’t corresponded to the highest perceived astringency as measured for Sagrantino and Nebbiolo (highest drying and lowest velvet), but similarly to these wines, Sangiovese were correlated to bitter taste and strong astringency sub-qualities (harsh, drying and dynamic), highlighted as discriminating sensory variables of the three wines compared to the others [5,6], showing that astringency is a very complex sensation to explain chemically [4,33,47–49]. It has been previously underlined that the structural characteristics of the flavan-3-ols play important role in astringency perception. For example, was demonstrated that for the same concentration catechin is less bitter than epicatechin [2], that galloylation is positively correlated to the “drying” astringent character, and epigallocatechin units tend to lower the “coarse” astringent perception [4].
L 349-: This paragraph should begin to make a nexus with the previous discussion. In the way it is, it seems unconnected of the previous discussion.
Done
L 355-357: A reference for this statement?
Done
L 355-368: It would be better to begin the sentence in a different way to better integrate it with the context.
Done
I consider that the Conclusions section could be improved with more factual statements drawn from the extensive and valuable results obtained by the authors.
The following paragraph was added:
In specific, the outputs of the study indicated that before phloroglucinolysis and in absolute values, Aglianico was the richest group in monomeric and dimeric fla-van-3-ols, while after the phloroglucinolysis it was Sagrantino the group showing the highest total flavan-3-ols amounts. Nerello Mascalese group showed the highest absolute values in polymeric trihydroxylated flavan-3-ols, indicating that is a variety rich in prodelphinidins. Sangiovese and Nebbiolo had the highest mDP values, and Corvina the lowest. Aglianico and Primitivo had the lowest percentages of B-ring trixydroxylated and C-ring trans conformation flavan-3-ols. The groups of Sangiovese, Montepulciano, Nerello Mascalese, and Teroldego were characterized by B-ring trihydroxylated flavan-3-ols (e.g., gallocatechin and epigallocatechin), and Nebbiolo and Corvina were the richest in galloylated analogues.
Reviewer 3 Report
This is a well-written and interesting paper about phenolic compounds in selected indigenous wines.
Every section is well-defined and described. I have no major remarks, but to add in M&M section all analyses that were done. For example, you do not mention mDP at all. If you have described it in a previous paper, you do not have to go into details but just mention all analyses that you have done. It seems like you did only phenols.
Author Response
This is a well-written and interesting paper about phenolic compounds in selected indigenous wines.
Every section is well-defined and described. I have no major remarks, but to add in M&M section all analyses that were done. For example, you do not mention mDP at all. If you have described it in a previous paper, you do not have to go into details but just mention all analyses that you have done. It seems like you did only phenols.
Dear reviewer, thanks for your positive feedback and the valuable comments.
Taking into account your comment, in the revised version of the manuscript the M&M section was enriched with further details, while information about the calculation are also given in the Supplementary Materials.
Round 2
Reviewer 1 Report
Line 177-181: only the abbreviations are needed, the full names already exist in the introduction (L.63-66).
Supplementary Materials : add reference for formulas for the calculations.
Author Response
Line 177-181: only the abbreviations are needed, the full names already exist in the introduction (L.63-66).
Done
Supplementary Materials: add reference for formulas for the calculations.
Done